# Dual-Gratings Imaging Spectrometer

**Rui Ouyang** [1,2], **Duo Wang** [1], **Longxu Jin** [1] **and Xingxiang Zhang** [1,*]

1   Changchun Institute of Optics, Fine Mechanics and Physics, Chinese Academy of Sciences, Changchun 130033, China; ouyangrui16@mails.ucas.edu.cn (R.O.); Wangduo@ciomp.ac.cn (D.W.); Jinlx@ciomp.ac.cn (L.J.)
2   Daheng College, University of Chinese Academy of Sciences, Beijing 100049, China
*   Correspondence: jan_zxx@163.com

**Abstract:** Common dispersive-type spectroscopic instruments include prism-type and grating-type, usually using a single dispersive element. The continuous imaging band is always limited by the dispersion angle. When it is necessary to image two wavebands with an ultra-spectral resolution that are far apart, the imaging is difficult due to the large diffraction angle. To broaden the spectral coverage of the imaging spectrometer, in this paper, we propose a dual-gratings imaging spectrometer with two independently rotating gratings. In this proposed system, two very far apart wavelength bands can be imaged in the adjacent areas by adjusting the angle of the dual gratings. This greatly expands the spectral coverage of the imaging spectrometer. Currently, the only application area considered for this instrument is solar applications. In this article, we present the optical system of the dual-gratings imaging spectrometer, illustrate several advantages of the new structure, and discuss new problems caused by the dual-gratings, which are referred to as overlap between two spectra and double image offset. We deduced the calculation process of the dual grating rotation angle, the relationship between the final acquired image and the slit, the relationship between the angle change between the dual gratings and the double image offset, and the relationship between the MTF upper limit reduction and the spatial frequency. This article also summarizes the shortcomings of this structure and studies the applicable fields under these shortcomings. At last, we simulate a dual-gratings imaging spectrometer system, compare this scheme with two traditional schemes, and conclude that this instrument has certain practical significance.

**Keywords:** spectrometer; spectral range; double gratings

## 1. Introduction

An imaging spectrometer is an instrument that is used in hyperspectral imaging and imaging spectroscopy to acquire a spectrally-resolved image of an object or scene [1]. It has been used in a variety of applications for specific target detection [2,3], precise classification [4–6], and the quantitative retrieval of biochemical or biophysical parameters [7–9]. Typical imaging spectrometers include HSI from America, ESA-HRIS and PRISM from ESA, COIS from US Navy's NEMO Program, FTHSI from MightySat II CHRIS on the European PROBA small satellite, and HYPERION on EO-1 satellite of America [10]. Nighttime light remote sensors that can capture imageries with high spectral and spatial resolution in a wide waveband and swath width were also investigated in [11]. In recent years, the need for high-performance imaging spectrometers that span an application space from satellite imagery to food safety has never been higher. In this data-driven world, consumers want to know more about the chemicals in their food or how to precisely match the paint color on their living room wall [12]. The spectral data with high spectral resolution that the imaging spectrometer can provide can be used to quickly and accurately identify the composition of the object, which has a high use value.

Sometimes, people are more concerned about the spectral images of discrete bands with ultra-spectral resolution. For example, in the process of flare outbreak, the most obvious visible bands are Hα (656.281 nm), H-line (396.8 nm), and K-line (393.4 nm) of ionized calcium. People often use the ultra-spectral solar surface images of these three bands for research. The traditional observation method uses filters. This scheme pays more attention to the spatial change of the monochromatic sun surface, and ignores the real-time subtle spectral changes in the waveband. If ultra-spectral observations in these three bands are required, according to the design method of the traditional planar grating imaging spectrometer, the spectral coverage is too large, and the diffraction angle is too large, which leads to difficulty in imaging and greatly reduces the image quality. Therefore, only two imaging spectrometers can be used to image the dual-band spectrum, which will increase the volume of the system. However, the dual-gratings imaging spectrometer can easily solve this problem.

The dual-gratings is an innovative structure that has been successfully used in many cases to improve the spectral resolution or spectral range of the system. In 2011, Kong Peng et al. studied a double-concave grating imaging spectrometer, which successfully improved the spectral resolution of the system [13]. In 2018, Liu studied a multi-grating spatial heterodyne spectrometer. By changing a single grating to multi-grating, interference images with high spectral resolution and large spectral range were successfully obtained [14]. In 2018, Xue researched a dual-grating dual-band imaging spectrometer, which divided the field of view into two and imaged different bands respectively [15].

As mentioned above, the dual gratings structure was used to improve the performance of the concave grating imaging spectrometer and the heterodyne interference imaging spectrometer. However, very few studies have applied it to planar grating imaging spectrometers. Xue's work uses this structure in a planar grating imaging spectrometer, but his structure divides the field of view into two parts and cannot be connected to common imaging systems. In this paper, this structure is applied to the planar grating imaging spectrometer to expand the system's spectral coverage and ensure high spectral resolution. Different from Xue's work, this research divides the aperture into two; each can image a waveband on the image plane, and can choose different spectral resolutions. Therefore, it has the potential to meet the requirements of wide spectral coverage and high spectral resolution at the same time, such as obtaining solar Hα line, ionized calcium H-line, and K-line ultra-spectral resolution images at the same time. Moreover, our system can be connected to common imaging systems and has the potential for conventional applications.

The principle of imaging spectrometer is shown in Figure 1. The spectral range is a very important parameter of the imaging spectrometer. Nevertheless, the band range is limited by the dispersion angle; hence, the continuous bands can be only partly imaged on the image plane. This new type of imaging spectrometer images the two far-apart bands on a single detector. Therefore, the spectral range can be divided into two halves that are far apart, thereby increasing the spectral coverage of the system. It further enables selecting different spectral resolutions according to the required signal-to-noise ratio. In this article, we show the above advantages using the experimental results. In our experiments, we further investigate the practical issues associated with this new structure and provide novel solutions in our experiments.

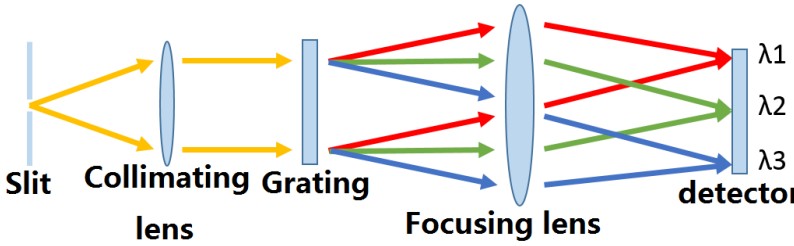

**Figure 1.** Principle of imaging spectrometer.

## 2. Dual-Gratings Imaging Spectrometer

### 2.1. Traditional Imaging Spectrometer

In general, a spectrometer is composed of a slit, collimating mirror, dispersing component, focusing mirror, and detector [16]. There are several classical imaging spectrometers, e.g., E-F structure, C-T structure, and Littrow structure. Here, we choose to use one of the simplest E-F structures as the basic structure for description, and other structures can also be selected. The traditional E-F structured spectrometers only use one mirror that serves both as collimator and imaging lens. A slit is also placed on the object plane. The light passing through the slit is then converted into collimated light by the mirror and exits at different dispersion angles after passing through the grating. It is finally imaged on the image plane by the same mirror as shown in Figure 2.

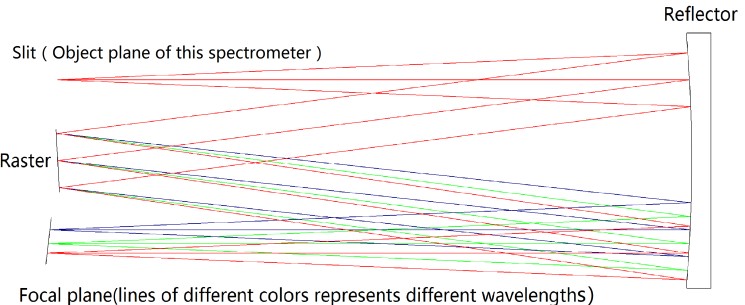

**Figure 2.** The E-F structure imaging spectrometer.

The main advantages of the E-F structure imaging spectrometer are its simple assembly and high-quality images.

### 2.2. The Role of Double Grating

In the traditional grating spectrometer, coma is eliminated by the symmetry of the system, so the image quality is higher at the image plane symmetrical to the slit. When the spectral coverage is too large, in the traditional grating system, the imaging area will inevitably be far away from the symmetrical area of the slit.

For example, as mentioned in the introduction, for solar hyperspectral imaging, the wavelengths are H$\alpha$ line, the H-line and K-line of ionized calcium, the focal length is 900 mm, the number of grating lines is 900, the slit is calculated at 8 microns, and the spectral resolution is up to for each waveband of 0.08 nm, the diffraction angles of the two wavebands are shown in the Figure 3.

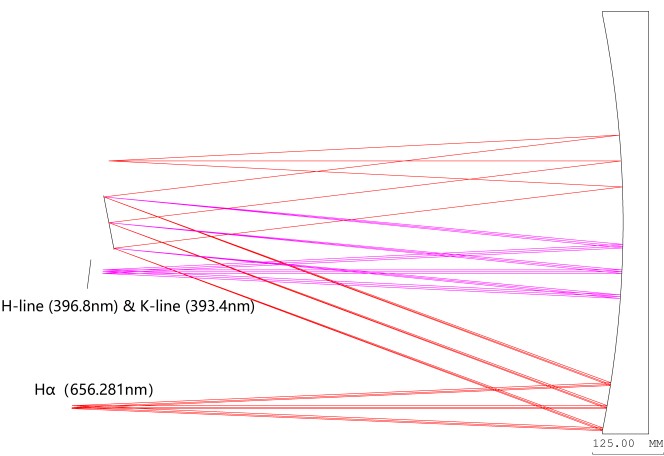

**Figure 3.** Traditonal method to design a spectrometer with H$\alpha$ line, H-line, and K-line of ionized calcium.

The imaging spectrometer of the traditional structure contains only one dispersive element, so it is constrained by the dispersion angle. As shown in the Figure 3, the H-line and K-line are imaged in the best image quality area, but the H$\alpha$ line has poor image quality because of the excessive diffraction angle. Therefore, in the traditional method, only two spectrometers can be used to image two bands respectively. However, this will increase the volume of the system.

The structure in Figure 4 changes the single grating to double grating, and the bands (H$\alpha$ line, H-line and K-line of ionized calcium) are concentrated in the same area by rotating the gratings. Therefore, the images of the two bands can be concentrated in the best image quality area, so the best image quality can be achieved in the whole waveband. The dual-gratings imaging spectrometer improves the spectral coverage on the basis of the single grating imaging spectrometer, eliminating the need to use two imaging spectrometers, and reducing the volume of the system.

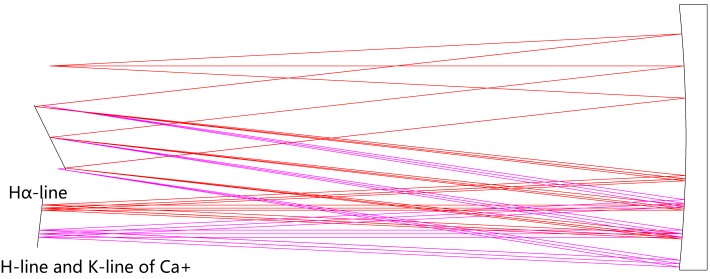

Hα-line

H-line and K-line of Ca+

**Figure 4.** Dual-gratings spectrometer with H$\alpha$ line, H-line, and K-line of ionized calcium.

### 2.3. Relative Position of Double Gratings

In order to better describe the relative position of the two gratings, a three-dimensional coordinate system (*x,y,z*) is established. The slit, two gratings, and spectral images are described in this three-dimensional coordinate system.

The slit is a line segment that passes (0,0,*h*)(0,−*a*,*h*), (0,*a*,*h*) and is parallel to the *y*-axis. The length of the slit is 2*a*, and the height relative to the center of the aperture is *h*. In order to simplify the description, the slit width of the slit is omitted here.

The grating 1 is a plane that passes (0,0,0)(0,*d*,0) line segment, which can rotate around the *y*-axis. The grooves of the grating 1 are parallel to the *y*-axis. The grating 2 is a plane that crosses the (0,0,0)(0,−*d*,0) line segment, which can rotate around the *y*-axis. The grooves of the grating 2 are parallel to the *y*-axis.The relative position of grating 1 and grating 2 is shown in Figure 10.

Regardless of the tilt of the image plane, the two bands of interest $(\lambda_1,\lambda_2)$ and $(\lambda_3,\lambda_4)$ can be imaged separately by rotating grating 1 and grating 2 around the *y*-axis. The imaging areas are *x* = 0, *y* ∈ (−*a*,+*a*), *z* ∈ (−*h*1,−*h*2) area and *x* = 0, *y* ∈ (−*a*,+*a*), *z* ∈ (−*h*3,−*h*4) area. The non-interesting bands will overlap with the interesting bands. For how to deal with the overlap problem, see Section 2.4, and for the size of the rotation angle of the gratings, see Section 2.5.

Figure 5 shows the position of the slit and the two spectra in the three-dimensional coordinate system.

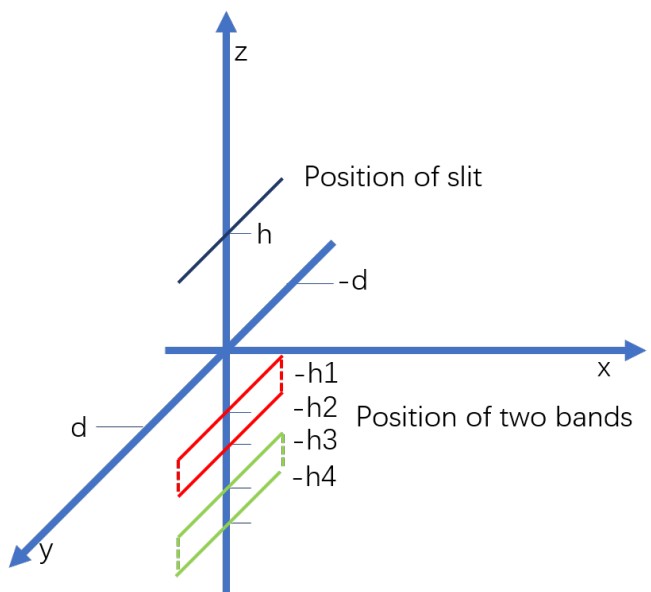

**Figure 5.** The position of the slit and the two spectra.

*2.4. Overlap between Two Spectra*

Overlap between two spectrums is a problem that needs to be paid attention to in dual-gratings imaging spectrometers. Compared with traditional imaging spectrometers, double gratings mean that there must be two overlapping energy bands. If the bands are not restricted, there must be spectral overlap. From the point of view of the generation of spectral overlap there are mainly two aspects. One is the overlap between the dual spectra produced by the dual gratings, and the other is the overlap between high-order spectra and imaging bands.

(1) For overlap between double spectra, this article has two ideas; the first one is the traditional method, adding filters in front of the gratings to prevent aliasing by constraining the imaging bands. The second is to use the response range of the detector. Imaging the second waveband in the unresponsive area of the first waveband can reduce one filter as shown in Figure 6. The second scheme is suitable for situations where the imaging band is close to the edge of the detector's response band.

(2) For the overlap between high-order spectrum and imaging bands, there is another way to be considered, that is, one grating works at +1 order, and the other grating works at −1 order. In this way, the high-order spectral imaging area can be staggered from the imaging area as shown in Figure 7.

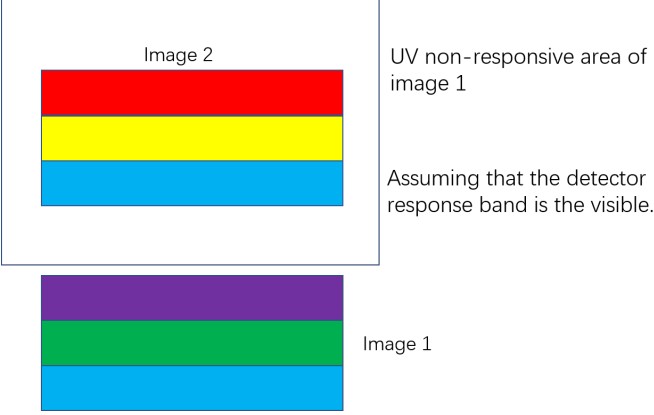

**Figure 6.** Image 2 is imaged in the unresponsive area in the ultraviolet band of Image 1, which can reduce one filter.

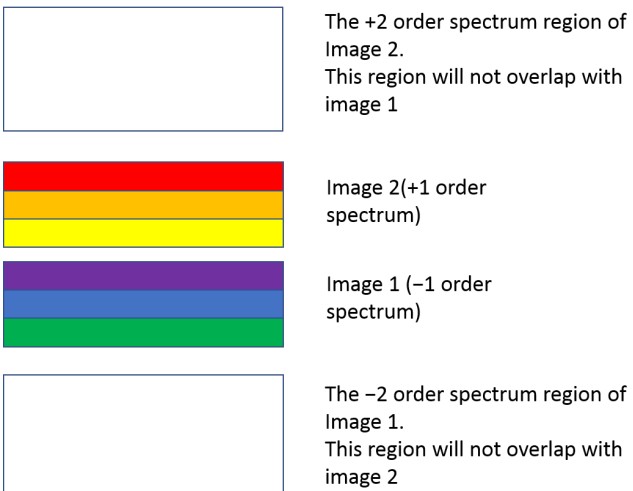

The +2 order spectrum region of Image 2.
This region will not overlap with image 1

Image 2(+1 order spectrum)

Image 1 (−1 order spectrum)

The −2 order spectrum region of Image 1.
This region will not overlap with image 2

**Figure 7.** Image 1 and image 2 adopt −1 order and +1 order respectively, which can prevent the influence of high-order spectrum.

When using the filters, the actual situation of the filter must be considered. The first case is that the center wavelength of the filter is usually shifted by a few nanometers. The second case is that usually the bandpass of the filter is expressed in the form of FWHM. However, in actual use, in order to avoid overlap between spectra, the bandpass whose response is less than 50% of the peak value should also be considered. For example, a piece of filter FB390-10 from the Thorlabs company. The center wavelength is 390 ± 2 nm and FWHM is 10 ± 2 nm. The transmission data is shown in the Figure 8. Assuming that the filter is applied to the above H-line and K-line, it can be seen from the data that within the drift of the center wavelength of ±2 nm, the H-line and K-line can be guaranteed to be within the response range. For the entire imaging plane, in order to ensure that the spectrum does not overlap, it is necessary to leave space for low response bands on the image plane. However, there is no need to consider the two edges of the filter's response band. Leaving imaging space for one of the edges allows the dual spectrum to not overlap.

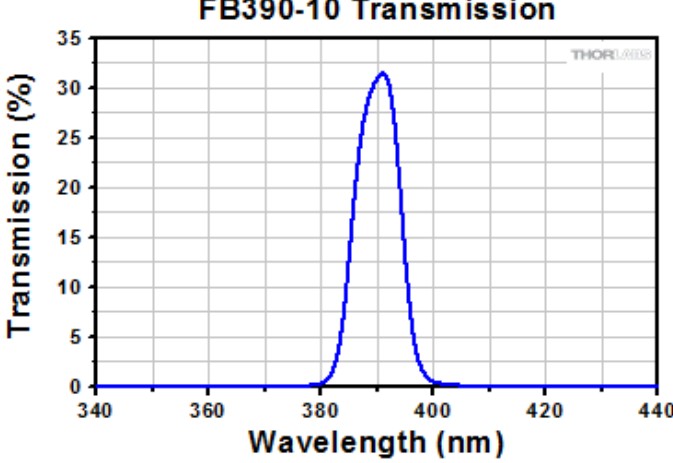

**Figure 8.** Transmission of FB390-10.

### 2.5. Rotation Angle of Double Gratings

Since the reflector of the imaging spectrometer of the E-F structure is a collimating lens and a focusing lens at the same time, the focal length of the reflector, the distance from the slit to the mirror, and the distance from the mirror to the sensor are all $s$. The grating is the aperture stop of the system. The distance from the slit to the center of the grating is $h$. The two imaging bands are $(\lambda_1, \lambda_2)$ and $(\lambda_3, \lambda_4)$. In the actual design, in order to

separate the two spectrums, the range of the two bands could be slightly expanded. It has been discussed above that using the +1-order and −1-order spectra of the two gratings respectively helps to eliminate the influence of high-order diffraction. Therefore, it can be assumed that $(\lambda_1, \lambda_2)$ comes from the +1 order spectrum, and $(\lambda_3, \lambda_4)$ comes from the −1 order spectrum. By rotating the two gratings to image $\lambda_1$ and $\lambda_3$ at one point, the sensor plane can be utilized to the maximum, and the angle of the gratings can be derived from the grating equation.

$\alpha$ is the incident angle of the grating, $\beta$ is the exit angle of the grating, and $\gamma$ is the angle of grating rotation (rotation to the slit is positive), as shown in Equation (1), where the first equation comes from the grating equation, and the second equation comes from the geometric derivation.

$$\begin{cases} sin(\alpha) \pm sin(\beta) = \dfrac{m\lambda}{d} \\ \alpha + \gamma = arctan(\dfrac{h}{s}) \end{cases} \tag{1}$$

In Equation (1), $m, d, \lambda, h, s$ are known numbers. The exit angle $\beta$ can be obtained by $\gamma$, or the rotation angle $\gamma$ can be inversely deduced by $\beta$.

Since $\lambda_1$ and $\lambda_3$ need to be imaged at the same point, in order to eliminate coma, this point can be positioned at the symmetry of the slit. According to the geometric relationship, we have Equation (2).

$$\beta - \gamma = arctan(\dfrac{h}{s}) \tag{2}$$

Through Equations (1) and (2), the grating rotation angle $\gamma$ can be obtained by bringing in $\lambda_1$ and $\lambda_3$, respectively.

$$\gamma_1 = arctan(\dfrac{h}{s}) - arcsin(\dfrac{\lambda_1}{2d_1}) \tag{3}$$

$$\gamma_2 = arctan(\dfrac{h}{s}) - arcsin(-\dfrac{\lambda_3}{2d_2}) \tag{4}$$

*2.6. Optical Structure of the Dual-Gratings Type Imaging Spectrometer*

The optical path of the dual-gratings imaging spectrometer is shown in Figure 9. The object surface is a slit.The light is converted into collimated light by the reflector, and after being dispersed by the double gratings, the two bands will be imaged in two adjacent areas. The dual-gratings are arranged in parallel along the reticle direction and can be used with suitable filters.On the image plane, the image has two dimensions, one is the spectral dimension and the other is the spatial dimension. The arrangement of the double grating is parallel to the space dimension, and the reticle is also parallel to the space dimension. This structure provides the following advantages: 1. Since the two gratings can be rotated independently, as long as the detector has sufficient response to the bands, two very far apart bands can be selected. 2. According to the needs of the signal-to-noise ratio, the gratings with different line pairs can be chosen to obtain two spectra with different spectral resolutions. In this system, to reduce the spherical aberrations, a parabolic mirror can be used to replace the traditional spherical mirror.

As introduced above, gratings with different line log numbers can be used according to the requirements of spectral resolution and signal-to-noise ratio. Figure 9 uses two gratings, 400 L/mm and 600 L/mm, to image the two wavebands $500 \pm 20$ nm and $400 \pm 15$ nm respectively. Since the number of line pairs per millimeter of the two gratings are different, the angle between the two gratings is small.

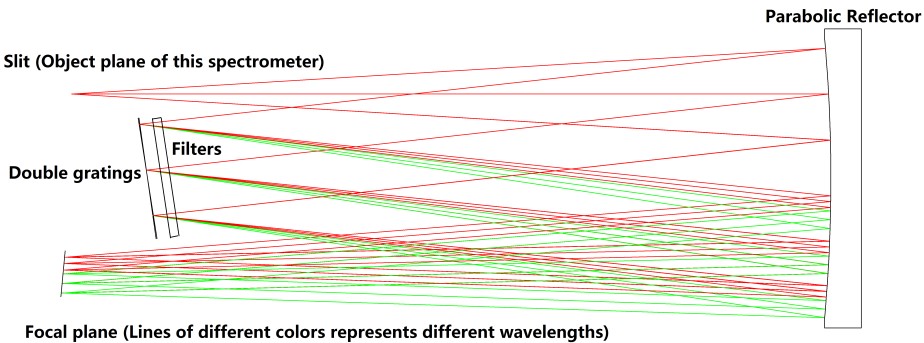

**Figure 9.** The dual-gratings imaging spectrometer.

### 2.7. The Relationship between the Slit and Image

Here, an SRF (spectral delta function) function is used to describe the spectral response of the spectral system. More detailed information about this method is in the article [17]. SRF is a parameter that only depends on the spectrometer. This parameter only traces the light from the slit to the image, which implicitly assumes that the light source directly illuminates the slit. Assuming that $x$ is the spatial dimension direction, $y$ is the spectral dimension direction, $y_0$ represents the width of the slit, $rect(y_0)$ is the convolution of the slit, $LSF_S$ is the $y$-line spread function of the spectral system, and $DET(y)$ is the pixel response of the detector. Therefore, Equation (5) can be obtained as:

$$SRF = rect(y_0) \otimes LSF_S \otimes DET(y) \tag{5}$$

Among them, $LSF_S$ can be analyzed in more detail. Here, $L1$ and $L2$ are the band position distributions of the two spectra. $L(y)$ is the response distribution of the imaging system, which does not consider the spectral factor. $F_1(\lambda)$ and $F_2(\lambda)$ are the response distribution of the filter to the wavelength, $G_1(\lambda)$ $G_2(\lambda)$ are the response distribution of the grating to the wavelength. $LSF_S$ is as shown in Equation (6) .

$$LSF_S = \frac{1}{2}[F_1(\lambda) \otimes G_1(\lambda) \otimes F_1(\lambda) \otimes L1 + F_2(\lambda) \otimes G_2(\lambda) \otimes F_2(\lambda) \otimes L2] \otimes L(y) \tag{6}$$

In Equation (6), $\frac{1}{2}$ represents the two bands occupying half of the energy respectively. Each of the bands needs to pass through the filter twice. The two spectra are separated by different spatial distributions $L1$ and $L2$. $L(y)$ represents the aberration effect of the system without considering the spectrum.

### 2.8. Double Images Offset Problem

Compared with the conventional imaging spectrometers, the dual gratings provide a wider spectral coverage and more free imaging methods. Nevertheless, the assembly errors might cause some unique issues, which are collectively referred to as the double images offset problem. By increasing or decreasing the angle between the dual gratings relative to the original design, the double image offset results in a relative change of the position and rotation angle of the two images. Severe double-image misalignment might make it impossible to image the double-image on the same detector, which will affect the spatial and spectral ranges of the image.

The angle between the double gratings is represented by the rotation angle $\alpha$ around the $x$-axis, the rotation angle $\beta$ around the $y$-axis, and the rotation angle $\gamma$ around the $z$-axis(see Figure 10). The impact of the double images offset problem can be analyzed according to the increase and decrease of the three angles relative to the original design (see, Table 1 and Figure 11).

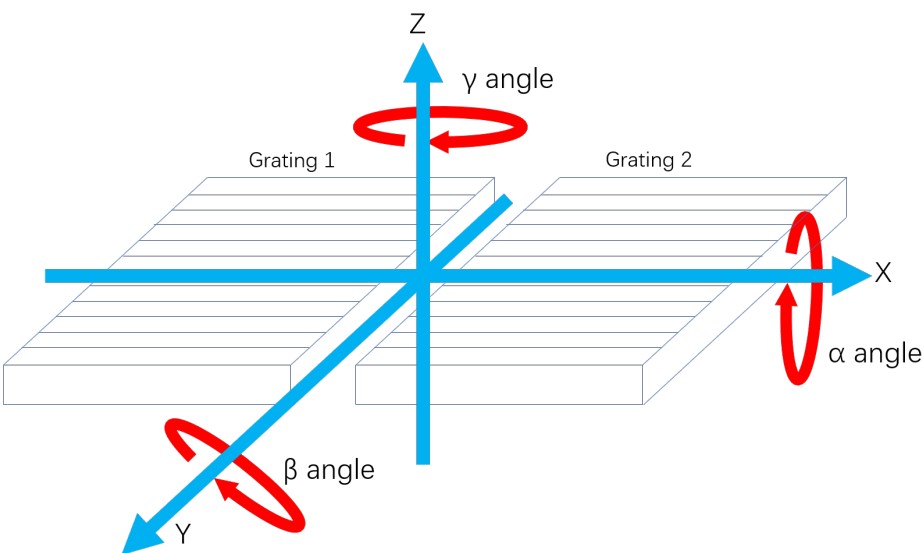

**Figure 10.** The direction of $\alpha$, $\beta$, and $\gamma$ angle.

**Table 1.** The Impact of the 3D angle variations.

| 3D Angle | Change | Influence |
|:---:|:---:|:---:|
| $\alpha$ | Increase | Distance between the two images expand |
| $\alpha$ | Decrease | Distance between the two images shrink |
| $\beta$ | Increase | Dislocation in the spatial dimension. |
| $\beta$ | Decrease | Opposite dislocation in the spatial dimension. |
| $\gamma$ | Increase | Tilt between the two images |
| $\gamma$ | Decrease | Opposite tilt between the two images |

Figure 11 is a schematic diagram of the double images offset problem. The red square and blue square each represent the image of a waveband on the image plane. The two images are parallel in the normal state, but due to the change of angle between the double gratings, misalignment and tilt appear between the two images.

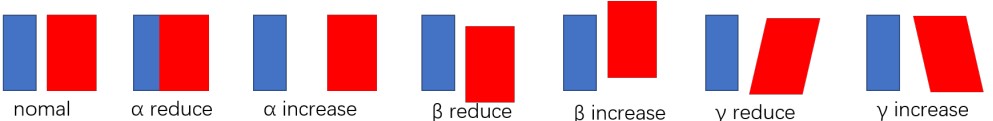

**Figure 11.** The impact of the double image offset (blue square and red square both represent one image).

The difference between the dual image offset problem and the tolerance lies in the difference in the measurement standards. The measurement standard for the tolerance is MTF (Modulation Transfer Function), and the dual images offset mainly affects the image position and tilt, which has only a slight effect on the MTF. Therefore, this issue needs to be considered based on system components, such as the adopted detector. For the given values of $\alpha$, and $\beta$, the number of pixels should be considered that are dislocated up, down, left, and right between the two images, which is acceptable at the detector. The value of $\gamma$, is also corrected as much as possible during the assembly and adjustment.

First, we derive the relationship between $\alpha$ rotation and dual image offset. From the above, the edge wavelengths $\lambda_1$ and $\lambda_3$ of the two bands are imaged at one point. By calculating the distance changed between $\lambda_1$ and $\lambda_3$ caused by $\alpha$ rotation, the relationship between the dual images offset and $\alpha$ can be obtained.

Assuming that the rotation angle between the dual gratings is increased by $t1$ relative to the original design, the distance between the dual images is increased by $h1$. The influence caused by the rotation $t1$ of the grating can be decomposed into two parts. The first part is the change of the incident angle caused by the rotation of the grating, which causes the change of the exit angle, and the second part is $t1$ which causes the reflected light to rotate as a whole by $t1$.

Here, the grating rotation angle change is recorded as $t1$, the exit angle change is recorded as $t2$, and the output light rotation angle is recorded as $t3$, which can be obtained from the Equation (7):

$$\begin{cases} sin(\alpha) \pm sin(\beta) = \dfrac{m\lambda}{d} \\ \alpha + \gamma = arctan(\dfrac{h}{s}) \\ sin(\alpha - t1) \pm sin(\beta - t2) = \dfrac{m\lambda}{d} \end{cases} \tag{7}$$

Among them, $h, s, m, \lambda, h, s, \gamma$ are known, and the formula can obtain the relationship between $t1$ and $t2$, that is, $t2$ can be obtained through $t1$. According to the two effects caused by $t1$, it can be determined that the relationship between the output light rotation angle $t3$, $t2$, and $t1$.

$$t3 = t1 + t2 \tag{8}$$

The rotation of the outgoing light causes the movement of the spot on the imaging lens. Since the system is an image-side telecentric system, the optical path from the focusing lens to the image plane does not participate in the bi-image shift. The optical path mainly involved in the dual image shift is the optical path from the grating to the mirror. Assuming that under the influence of $t1$, the change in the height of the reflected light is $h1$. Due to the complex calculation process, it is approximated here and deduced from the geometric relationship. The relationship between $h1$ and $t3$ is shown in Equation (9).

$$h1 \approx h \times (\frac{sin(\beta - \gamma + t3)}{sin(\beta - \gamma)} - 1) \tag{9}$$

The effect of $\beta$ rotation on the double image offset is relatively simple. Due to the distribution of double gratings, $\beta$ rotation is similar to the rotation of a plane mirror. Assuming that there is $\beta$ rotation $k1$, the exit angle increases by $k2$, and the exit light rotation angle is recorded as $k3$. Then, there is $k3 = k1 + k2 = 2k1$. Assuming that under the influence of $k1$, the change in the height of the reflected light is $h2$. By geometric relations, the relationship between $h2$ and $k1$ is shown in Equation (10).

$$h2 \approx d \times 2sin(k1) \tag{10}$$

The effect of $\gamma$ rotation on the double images offset is also relatively simple. The rotation of $\gamma$ only affects the direction of dispersion, so the relative rotation of the grating by m1, the rotation of the dispersion direction will also rotate with $m1$.

### 2.9. The Effect of Double-Gratings on MTF

Because the plane grating used in this article does not have focal power, changing the traditional single-plane grating to a dual-plane grating has no effect on the system aberrations. The biggest impact of the dual-gratings on the image quality of the optical system should be the impact on the diffraction limit. The diffraction limit of an optical system is shown in Equation (11).

$$A = 1.22f\frac{\lambda}{D} \tag{11}$$

where $f$ is the focal length of the system, $D$ is the diameter of the entrance pupil of the system, and $\lambda$ is the wavelength. Due to the use of dual-gratings, only half of the aperture in each band is involved in imaging, that is, $D$ is half of the single grating. Therefore, its



diffraction limit will be doubled, resulting in the MTF cut-off frequency being reduced to half, which will cause The reduction of MTF. The amount of decrease of the upper limit of the MTF can be derived.

Then the upper limit of MTF can be calculated as a formula, where $x$ is the spatial frequency, and $y$ is the upper limit of MTF:

$$y = -1.22f\frac{\lambda}{D} \times x + 1 \tag{12}$$

The upper limit of MTF for reducing the pore size can be calculated as:

$$y = -1.22 \times \sqrt{2} \times f\frac{\lambda}{D} \times x + 1 \tag{13}$$

Therefore, the MTF reduction varies with spatial frequency as:

$$\delta y = 1.22 \times (\sqrt{2} - 1)f\frac{\lambda}{D} \times x \tag{14}$$

*2.10. Limitations of Dual-Grating Imaging Spectrometers*

The dual-grating imaging spectrometer can acquire two bands that are far apart. However, compared with the traditional single-grating imaging spectrometer, the current dual-gratings imaging spectrometer has the following shortcomings:

(1)  Cost increase. In order to prevent overlapping of spectra, the dual grating imaging spectrometer needs to add two filters and one grating, which increases the cost.

(2)  The low energy utilization rate. This shortcoming is composed of two reasons. The first is because the double gratings bisects the aperture into an image, resulting in only half of the total energy in each band. The second reason is because in the imaging process, it needs to pass through the filter twice, which reduces the energy utilization rate.

(3)  Stray light is more complicated. Due to the lower energy utilization rate of dual gratings, the stray light in the system is more than that of traditional single gratings.

Considering these limitations, the current structure should be used in fields that are abundant in energy, insensitive to cost, and have ultra-spectral imaging requirements for multiple bands with large intervals. Therefore, this article suggests the solar application field.

## 3. Simulation of a Dual-Gratings Spectrometer

Next, we design a dual-grating imaging spectrometer for H$\alpha$, ionized calcium H-line and K-line hyperspectral imaging. Here, an optical design software CodeV is used to simulate the system. The focal length of the spectrometer was 600 mm, the number of pairs of double grating lines was 1350 L/mm, the numerical aperture of the object side was 0.05, and the size of the slit was 16 mm × 5 μm. The double gratings work at +1 and −1 order to avoid the influence of the high-order diffracted light of the gratings. Changing one reflector of the E-F system to two can better eliminate aberrations.This can correct coma in an asymmetrical state. This correction method comes from Shafer's research [18]. In the simulation, in order to make the simulation closer to reality, data from two filters FB390-10 and FB650-10 from Thorlabs were used. The bandpass of the two filters are 380–400 nm and 640–660 nm, respectively. Therefore, it is only necessary to leave 396 nm–400 nm and 656 nm–660 nm imaging space between the two bands, and the dual bands will not overlap. The light path is shown in the Figure 12 and the system parameters are shown in Table 2.

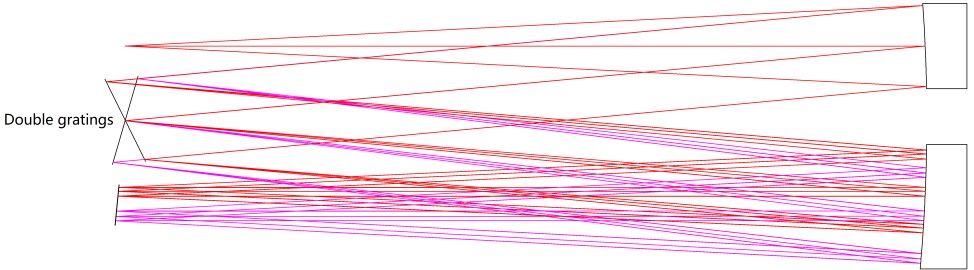

**Figure 12.** Optical structure of dual-gratings spectrometer.

**Table 2.** Parameters of the experimental system.

| Parameter | Value |
|---|---|
| System focal length | 600 mm |
| Waveband | 391.4 nm–398.8 nm, and 653.281 nm–658.281 nm |
| Grating line pairs | 1350 L/mm and 1350 L/mm |
| Slit | 16 mm × 5 μm |
| Detector | 5 μm (To calculate the number of spectrum data) |
| NA(object side) | 0.05 |

## 4. Results

Through software simulation, we obtained the data as shown in Table 3.

**Table 3.** Data obtained through simulation.

| Data | Value |
|---|---|
| Spectral resolution | Hα each band 0.005 nm & H and K each band 0.006 nm |
| MTF in 32 L/mm | higher than 0.6 |
| Smile | Hα is 3.5 μm & H and K is 3.7 μm |
| The volume of the spectrometer | 0.015 m$^3$ |
| Number of output spectrum data | Hα 1000 & H and K 1234 |
| Grating widths | 33 mm × 66 mm × 2 |
| Mirror size | 60 mm × 60 mm and 60 mm × 88 mm |
| Angle between two gratings | 42.85° |

Because the double gratings work in the +1 order and −1 order, the angle between the double gratings is relatively large, as shown in the Figure 12.

According to the data in Table 3, the MTF (modulation transfer function) of the full field-of-view system is higher than 0.6 at 32 L/mm, indicating that the system image quality is good. The spectral resolution is as high as 0.005 and 0.006 for each band, which is enough to meet the observation requirements of ultra-spectral resolution. The system can simultaneously output 1000 spectrum data near the Hα band and 1234 spectrum data near the H-line and K-line of ionized calcium.

In the case of using the same grating, two imaging spectrometers with the same spectral resolution designed according to the traditional scheme can obtain the nearly same parameters as the dual-grating system, and the system volume can be 0.028 m$^3$ according to the simulation, which is almost twice that of the dual-grating system.

## 5. Comparison with Traditional Scheme

Here, we are still taking the ultra-spectral imaging with Hα line, ionized calcium H-line, and K-line of sun as an example. For the spectrum observation of the sun's surface in the three bands, the most mainstream solution for ground-based telescopes is to use filters, and some also use birefringent filters or polarization spectrometers [19]. These schemes all take images of the sun in a very narrow band. Therefore, more attention is paid to temporal

and spatial changes, ignoring continuous changes in the spectrum. Solar observation satellites generally choose ultraviolet or extreme ultraviolet bands, and only analyze the visible light band as a whole. In these solutions, the filters can provide a very narrow band of high-precision solar imaging, so it is most widely used. However, it cannot provide continuous spectral data unless a new filter is replaced. Grating-type imaging spectrometer is an option commonly used to provide real-time continuous spectrum data. At present, traditional planar grating imaging spectrometers are used in some solar ultra-spectral imaging projects, such as the scientific research project 'Chinese Hα Solar Explorer'.

Tables 4 and 5 take the ultra-spectral imaging of these three bands as an example to compare traditional imaging spectrometer, the filters and the dual-gratings imaging spectrometer.

**Table 4.** Comparison of Traditional imaging spectrometer and Dual-gratings scheme.

|  | Dual-Gratings Spectrometer | Traditional Spectrometer |
|---|---|---|
| Output image | 2-D images (spectral and spatial) | 2-D images (spectral and spatial) |
| Spectral resolution | high | high |
| volume | small | big |
| suitable for satellite | Yes | Yes |
| Currently adopted | rare | rare |

**Table 5.** Comparison of filter and Dual-gratings scheme.

|  | Dual-Gratings Spectrometer | Filter |
|---|---|---|
| Output image | 2-D images (spectral and spatial) | Monochrome image |
| Spectral resolution | high | low |
| the acquisition speed | low (need time to scan) | real time |
| Features | Continuous spectrum data | High-precision spatial information |
| Number of spectrum data | Multiple | 1 (Adjustable ) |
| Currently adopted | Rare | Broadly used |

Compared with the traditional imaging spectrometer and the dual-gratings imaging spectrometer, the dual-gratings imaging spectrometer can obtain high spectral resolution while the volume is much smaller than the traditional design.

Compared with the filter and dual-gratings scheme, the filter has high spatial resolution and can obtain high-precision solar images, but cannot output continuous spectrum data in real time. In 2008, in the TESIS experiment, two telescopes (FET) used filters with mirror coatings to obtain solar images with a bandpass of 0.6 nm [20]. However, this data is still low compared to imaging spectrometers. The dual gratings spectrometer can output thousands of continuous spectrum data in real time.This makes it possible to have more discoveries in the spectral dimension.

## 6. Conclusions

Our main objective is to address the limited spectral range of the conventional imaging spectrometers, which are limited by the dispersion angle. We propose an innovative structure, namely, dual-gratings imaging spectrometer.The instrument can acquire ultra-spectral resolution spectral images for two wavebands, and has the potential to be applied to the sun's ultra-spectral imaging. This article introduces the structure of traditional imaging spectrometers, presents the advantages of dual-gratings and analyzes the practical issues caused by the double gratings-overlap of two spectra and double image offset problem. This paper studies the calculation process of dual grating rotation angle, the relationship between the image and the slit, the relationship between the angle between the dual gratings and the dual image offset, and the relationship between the reduction of the upper limit of MTF and the spatial frequency. This article also summarizes the shortcomings of this

structure and studies the applicable fields under these shortcomings. At last, we simulated a dual-gratings imaging spectrometer with optical design software and compared it with two traditional schemes. Compared with the birefringent filter, the dual-gratings imaging spectrometer can provide real-time continuous spectrum data. This gives more opportunities for discoveries in the spectral dimension. Compared with traditional imaging spectrometers, the dual-gratings imaging spectrometer has the same spectral resolution and spatial resolution, and is significantly smaller in volume.

**Author Contributions:** Conceptualization, R.O. and X.Z.; methodology, D.W. and R.O.; software, R.O.; validation, R.O. and D.W.; formal analysis, R.O. and L.J.; investigation, X.Z.; resources, L.J.; data curation, R.O.; writing—original draft preparation, R.O.; writing—review and editing, X.Z.; visualization, R.O.; supervision, X.Z. and L.J.; project administration, R.O.; funding acquisition, X.Z. All authors have read and agreed to the published version of the manuscript.

**Funding:** This research received no external funding.

**Institutional Review Board Statement:** Not applicable.

**Informed Consent Statement:** Not applicable.

**Acknowledgments:** The authors thank Tianjiao Fu for reimbursement of the funds, which significantly helped in this research.

**Conflicts of Interest:** The authors declare no conflict of interest.

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
