# Peer review of "Dual-Gratings Imaging Spectrometer"

_applsci, doi:10.3390/app112211048_

Round 1

Reviewer 1 Report

There are major concerns about this work:

1) It seems that you apply your proposed design only for solar application. If it is the only field you plan to utilize your two-grating spectrometer you have to clearly specify it in the title and abstract, otherwise you have to assume a more general point of view, especially referring to the comparison with other layouts (Tables 2 and 3) and Conclusions.

2) From Figs. 7 and 9 the spectral bands on the detector seem placed side by side with respect to the dispersion plane, that is the segment that links the centers of the two spectral band is on the dispersion plane. It is correct? However, you have to specify their position.

3) If the spectral bands on the detector are placed side-by-side, you must explain how you avoid that the diffracted light from a grating (out of the band of interest) overlaps on the detector the band produced by the other grating. Maybe you could utilize two interferential filters (the filters cited at line 120?), but you have to describe their position and characteristics, taking in the account the actual rejection level in order to verify their effectiveness.

4) You must extensively describe the drawbacks of your proposal and the limits in its applicability: less energy, stray light, less resolving power, cost...

Other requested revisions in the attached file.

Author Response

Hello, thank you very much for your comments, I did not figure out how to insert picture, so I put the reply in the attachment. 

Reviewer 2 Report

One variant of a spectrometer having two gratings is described in the article. The article does not present a detailed theory of a two-gratings spectrometers, where the basic relations for calculating the image of the input slit would be derived. Furthermore, the calculation of the effect of aberrations of the imaging system (mirror) on the image of the input slit is completely missing. Figures 4 and 6 are completely confusing and must be redrawn in order to give the reader a clear overview of the optical system of the spectrometer. Equation (1) is given without derivation and indication of the limits of its validity. The literature mentioned in the article is also insufficient. The article in this form does not provide for the reader the information needed to design his own spectrometer.

I recommend that the article be substantially rewrite and supplemented with the necessary theory and literature.

Author Response

Hi, thank you very much for your comments. This is the case. This article converts a traditional single-plane grating to dual-plane gratings. Since the gratings are flat, this conversion does not change aberration in the original system. The aberration depends more on the choice of other components and the actual detector, so it is not the focus of this article. Since the article converts a single grating to double gratings, it pays more attention to the changes caused by double gratings structure. Thanks again for your comment.

Round 2

Reviewer 1 Report

1) Change the sentence at rows 29-32 because now it is a nearly exact citation of ref. [12]

2) From both the Figs and your answer, I have understood that the two grating are superimposed (with respect to the instrument plane as shown in Fig. 8), that is a grating is above the other, with their grooves in vertical direction. In this case, the images on the detector should be too superimposed if the angle beta is 0, that is an image is above the other. It is correct? However, you MUST improve the description of the design and clearly show the position of the grating and the images that they generate.  Maybe it could be useful to utilize the same reference system for all the Figures and show it in them.

3) It is not clear if the images on the detector are side-by-side or superimposed with respect to the dispersion angle. Please better specify that, because it is very important with respect to the problem of the overlapping spectra. Obviously the spectra position must be coherent with the gratings position and orientation.

4) At row 92 substitute "parallel light" with "collimated beam".

4) In Fig. 9 substitute "direction" by "angle"

5) Please carefully check language details (upper/lowercase, space after commas and full stops, ...); substitute "I think..." at row 286 with an impersonal construction.

Author Response

Response to Reviewers

[Cover Letter]

Dear Editor,

We appreciate you and the reviewers for your precious time in reviewing our paper and providing valuable comments. It was your valuable and insightful comments that led to possible improvements in the current version. The authors have carefully considered the comments and tried our best to address every one of them. We hope the manuscript after careful revisions meet your high standards. The authors welcome further constructive comments if any.

Below we provide the point-by-point responses. All modifications in the manuscript have been highlighted.

Sincerely,

Authors Rui Ouyang, Duo Wang, Longxu Jin and Xingxiang Zhang

[Minor Comment 1] Change the sentence at rows 29-32 because now it is a nearly exact citation of ref. [12]

Response: It has been corrected. Thanks a lot.

[Minor Comment 2]From both the Figs and your answer, I have understood that the two grating are superimposed (with respect to the instrument plane as shown in Fig. 8), that is a grating is above the other, with their grooves in vertical direction. In this case, the images on the detector should be too superimposed if the angle beta is 0, that is an image is above the other. It is correct? However, you MUST improve the description of the design and clearly show the position of the grating and the images that they generate.  Maybe it could be useful to utilize the same reference system for all the Figures and show it in them.

Response: Thank you for your valuable comments. I added a subsection and established a three-dimensional coordinate system. The positions of the slit, gratings, and two-bands images are described in this coordinate system. I believe this will make the relationship between them clearer.

[Minor Comment 3] It is not clear if the images on the detector are side-by-side or superimposed with respect to the dispersion angle. Please better specify that, because it is very important with respect to the problem of the overlapping spectra. Obviously the spectra position must be coherent with the gratings position and orientation.

Response: I added a picture, Figure 5, to describe the positional relationship of the images, hoping to make the relative positions of the images clearer.

[Minor Comment 4] At row 92 substitute "parallel light" with "collimated beam".

Response: It has been corrected. Thanks a lot.

[Minor Comment 5] In Fig. 9 substitute "direction" by "angle"

Response: It has been corrected. Thanks a lot.

5) Please carefully check language details (upper/lowercase, space after commas and full stops, ...); substitute "I think..." at row 286 with an impersonal construction.

Response: The details have been checked, thank you very much.

【Modified part of the text】

In the section 'introduction' , I modified the last sentence of the first paragraph and put all the quotes before the period.

In the section 'Dual-gratings imaging spectrometer', I modified ‘parallel’ to ‘collimated’, and added a subsection and a picture (Figure 5) to describe the positional relationship between slits, gratings and images. Corrected some spelling errors. Added some descriptions of Figure 9. Change the ‘direction’ of Figure 10 to ‘angle’.

Reviewer 2 Report

No comments

Author Response

Thank you very much for your kind comments.

This manuscript is a resubmission of an earlier submission. The following is a list of the peer review reports and author responses from that submission.

Round 1

Reviewer 1 Report

This paper presents the scheme of a dual-gratings imaging spectrometer to expand the spectral coverage of the instrument. The introduction states the purpose of the paper, and the relation between the paper and previous works is clearly explained. The feasibility of the presented scheme is illustrated via an experiment based on a double-grating imaging spectrometer model.

The following are my specific comments:

(1) In the presented scheme, two far apart wavelength bands can be imaged in the same detector by adjusting the angle of the dual gratings. I wonder whether it is necessary to image the far apart wavelength bands in the same detector. In addition, how is the angle of the dual gratings adjusted? Is it adjusted manually or automatically? It is better to give some explanations.

(2) Are there any difference between “e-f structure” and “E-F structure” in Section 2?

(3) The full names of “MTF” and other abbreviation in the paper should be given right after they first appear.

(4) The authors are suggested to summarize the main strategies to optimize the optical design of the dual grating imaging spectrometer. Try to provide some approaches to analyze the potential performance of the instrument theoretically.

(5) It is better to provide some performance comparison between the presented scheme and other existing methods via simulation or experiment.

(6) In Section 7, there may be a typo error in the sentence “Figure 11 is the spectrum from Figure 11”.

(7) The experimental results are not satisfactory. It seems that Figure 11 is ambiguous. How is the image processed? It is better to provide a brief specification about the method. There are no labels in both axes of Figure 12. In addition, are there any numerical indexes to evaluate the spectrum coverage and the spectral resolution?

(8) It is stated in Section 8 that dual-gratings imaging spectrometer is a field with great potential. The authors are suggested to provide an explicit description for a potential application instance.

Reviewer 2 Report

Review of manuscript entitled “Dual-Gratings imaging spectrometer” by Rui Ouyang et al.

The spectral coverage of an imaging spectroscope is enlarged by the use of two gratings. The optical system is presented and some commentaries and details are given.

The introduction gives a view of other attempts made to increase the spectral coverage. Then the suggested two-grating spectrometer is presented. Parabolic mirrors are used to reduce the system aberrations. Two gratings were chosen. In the manuscript line 98 it is said “To reflect the advantages in the previous article,….” it is not mentioned a reference so it is not know which article it is referred to.

Section 4 deals with what is called the Double images offset problem. Some angles are described but it would be good presenting a drawing that shows the angles. The paragraph after Table 1 is confuse. The MTF is mentioned and a formula is given. Mention how did this formula was obtained. Section 5 also need some good explanation, perhaps based on a drawing.

In section 6 is explained the experimental set up. It is mentioned that a filter is placed in front of just a grating. However, in the Figure 7 the filter covers the two gratings. In Table 2 some characteristics of the system are shown, What is the meaning of “Detector”, visible 6.9 microns?. How did you achieve the Spectral resolution of 23.9 nm and 15.88 nm?. What is the meaning of “Lines curved”. How did you get the NA of 0.05

Photographs 8, 9, 10, 11 are very bad. When reporting experiments photographs are not a good way to show the set up. It is better to show figures or schemes. Photographs support these schemes. If words are inserted in the photographs they should be clear and the use of arrows is suggested. Try to use a laboratory to make the experiments. By looking the photographs it seems the experiments were done over a desk.

Paragraph after Fig. 10 does not explain well the ideas behind.

Finally, some references, for example 13, 14 and 15 are not well written.

Reviewer 3 Report

Manuscript Number: 1377580

Title: Dual-Gratings Imaging Spectrometer

Authors: Rui OuYang, Duo Wang, Longxu Jin, Xingxiang Zhang

The manuscript by R. OuYang et al. presents experimental realization of the dual-grating spectrometer. The chance to develop a planar grating imaging spectrometer for simultaneous detection in two spectral intervals is impressive. The authors make a good effort in trying to develop the new tool which can be connected to common imaging systems. Overall, the results reported are clear. The problem is that although there are important achievements in spectrometer design the text of the paper needs significant improvement. Unfortunately, I cann’t recommend this paper for publication in it’s current form as it does not meet the quality requirements for a scientific manuscript.

The following points should be addressed for improving the paper:

  1. In my view, the Abstract is vague. The piece of information presented (lines 1−5) seems redundant for the Abstract. Moreover, it is rather for readers who are far from the problem.
  2. Introduction:

Some sentences in the Introduction are taken word for word from other sources (page 1, lines 17–18; lines 26–31). In this case, they should be taken in quotation marks.

  1. The authors claim in Introduction that: «Due to the limited CMOS image elements, the wide spectral range and high spectral resolution cannot be achieved at the same time» (Page 1, lines 33–34). It is difficult to agree with this statement since the modern level of technology makes it possible to reach high resolution of spectrometers in a wide range. (For example, CCD detectors can be used in the system). On the other hand, it is hard to achieve high spectral resolution and high spatial resolution at the same time.
  2. The discussion (page 8, lines 170-174) is weak. To my mind, there is no any in-depth discussion.
  3. Table 2:

The value of the spectral resolution is indicated as 23.9 nm/mm for λ=400–440 nm and 15.88 nm/mm for λ=485 – 525 nm. But the dispersion has the dimension “nm/mm”, while the spectral resolution is indicated in nm for each spectral range.

  1. Page 8, line 172: “Therefore it enables simultaneous detection of the ultraviolet and infrared bands”.

However, the infrared region of the spectrum begins at 760 nm. In Table 2, the system wave bands of 400 nm–440nm and 485 nm–525nm are indicated only. In Fig. 12 the long-wavelength threshold of the spectrum is 700 nm. Perhaps, the authors meant the ultraviolet and visible ranges?

  1. There are inaccuracies in the manuscript:

- Fig. 8, Fig. 9:

The text on the figures themselves is poorly readable. The text on the Fig. 9 completely repeats the figure caption.

- Fig.8 and Fig. 10 have the same captions and seem to be interchangeable.

- Fig. 12:

1) The caption The spectrum from the Figure 10” is unclear. In Fig. 10 we see a photo of the experimental system. What did the authors mean?

2) There are no the axis titles at all.

  1. It would be useful to give the representation of the spectrum actually registered by the suggested spectrometer for identification of any substances.
  2. The English of the paper needs to be significantly improved.
